# Does Reasoning Improve Seeing?
# Understanding When Vision-Language Models Benefit from Thinking

**Jing Bi** [1]  **Luchuan Song** [1]  **Dingxin Zhang** [2]  **Pinxin Liu** [1]  **Guangyu Sun** [3]  **Lianggong Bruce Wen** [4]  **Weidong Cai** [2]  **Chen Chen** [3]  **Chenliang Xu** [1]

## Abstract

Vision-language models (VLMs) now support both direct Instruct and explicit-reasoning Thinking modes, yet practitioners still lack principled ways to decide when reasoning actually improves performance, or how much computation to spend at test time, so we investigate whether VLMs encode meta-cognitive signals for adaptive inference. We derive oracle labels for two properties: (1) reasoning helpfulness, namely whether explicit reasoning improves accuracy, and (2) desired generation length, the minimal token budget needed for a correct answer. Probing final-layer representations in InternVL and Qwen3-VL models, we find Thinking models encode these signals more linearly than Instruct models, suggesting that reasoning-oriented training enhances meta-cognitive structure. Head-wise attribution reveals two circuits: length-control heads in lower layers and reasoning/difficulty heads in higher layers. Causal interventions confirm these roles across benchmarks: scaling length heads controls output length with minimal accuracy loss, while scaling reasoning heads enables a perception-reasoning trade-off, improving accuracy by up to 5.3%. Our results demonstrate that reasoning-tuned VLMs develop localized, manipulable circuits for meta-cognitive control, enabling test-time steering of computation and reasoning without retraining.

[1]Department of Computer Science, University of Rochester, Rochester, NY, USA [2]School of Computer Science, The University of Sydney, Sydney, NSW, Australia [3]Center for Research in Computer Vision, University of Central Florida, Orlando, FL, USA [4]Corning Incorporated, Corning, NY, USA. Correspondence to: Jing Bi <jing.bi@rochester.edu>.

*Proceedings of the 43$^{rd}$ International Conference on Machine Learning*, Seoul, South Korea. PMLR 306, 2026. Copyright 2026 by the author(s).

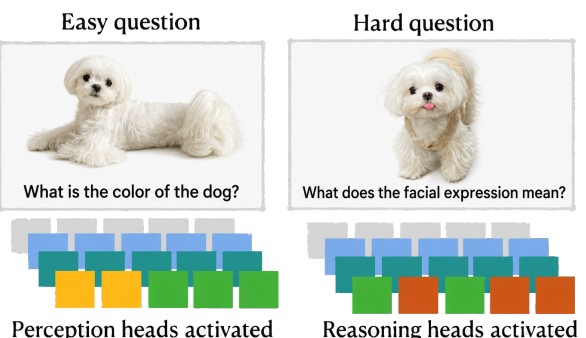

*Figure 1.* **Difficulty-dependent head activation.** Easy questions induce minimal changes across attention heads, whereas difficult questions selectively activate deeper-layer heads, encoding the need for additional reasoning.

## 1. Introduction

Large vision-language models (VLMs) are increasingly evaluated on problems that require both accurate visual perception and multi-step reasoning. Benchmarks such as MMMU stress this combination by pairing heterogeneous images (charts, diagrams, scientific figures) with expert-level questions across many disciplines (Yue et al., 2024). In response, modern VLM families expose multiple inference modes: a concise INSTRUCT mode that answers directly, and a THINKING mode that emits explicit reasoning traces before the final answer. A practical but unresolved question is how to *adaptively allocate* test-time computation: when does explicit reasoning improve versus degrade performance, and what is the desired generation length?

Recent evidence shows that more reasoning can actually hurt in multimodal settings. Bi et al. find that long reasoning trajectories tend to drift away from visual grounding, producing wrong answers despite the extra reasoning, and they recover performance with uncertainty-guided lookback prompts that re-anchor the model to the image (Bi et al., 2026). This motivates instance-wise control policies that determine *when* to reason and *how* to maintain grounding, rather than relying on fixed token budgets or always-on chain-of-thought.

In language-only settings, lightweight signals have emerged for test-time compute allocation. Confidence-based meth-

ods filter unproductive reasoning traces (Fu et al., 2025), while Ni et al. show that small auxiliary heads trained on frozen internal states can verify reasoning quality (Ni et al., 2026). Long-horizon evaluations reveal that reasoning models struggle to budget computation effectively as compositional dependencies grow (Lu et al., 2026), reinforcing the need for input-adaptive controllers.

A key insight from mechanistic interpretability is that models may represent properties such as difficulty or reasoning utility before generating an answer. Lee et al. show that problem difficulty is linearly decodable from final-token representations in LLMs and can be localized to specific attention heads, enabling causal interventions through head-output scaling (Lee et al., 2025). This motivates a natural extension to VLMs: rather than treating reasoning as a binary switch, we identify internal signals that predict *when reasoning helps* and expose localized circuits that can be steered toward perception or reasoning behavior.

We investigate whether VLMs internally encode two meta-cognitive properties critical for adaptive inference: (1) *reasoning helpfulness* $r_{\text{help}}$, whether Thinking mode improves accuracy over Instruct mode on a given example, and (2) *desired generation length* $\ell_{\text{opt}}$, the token budget that achieves correctness efficiently. We obtain supervision via a multi-pass evaluation procedure that runs $K=10$ stochastic generations in both modes, estimating correctness rates and selecting the winning mode and its shortest correct response as the oracle length.

Using final-layer last-token embeddings as our sole input, we train linear probes to predict these targets and investigate three questions:

**Q1 (Experiment 1):** Are these meta-cognitive signals linearly decodable, and do Thinking models encode them more explicitly than Instruct models?

**Q2 (Experiment 2):** Which attention heads are causally responsible for these signals? Are length control and reasoning assessment computed by distinct or overlapping circuits?

**Q3 (Experiment 3):** Can we steer model behavior, controlling generation length or shifting between perception and reasoning, via interventions on identified head sets?

Across InternVL and Qwen3-VL families, we find:

**Thinking models encode meta-cognition more linearly** (Exp. 1): Probes on Thinking representations achieve 29% lower MSE for $r_{\text{help}}$ and 18% lower MSE for $\ell_{\text{opt}}$ compared to Instruct probes, indicating that reasoning training reorganizes representations.

**Length and reasoning circuits are partially disentangled** (Exp. 2): Length-control heads concentrate in middle-to-lower layers (mean layer 15), while reasoning/difficulty

heads cluster in higher layers (mean layer 27) with 0.68–0.76 Jaccard overlap, suggesting a shared difficulty-assessment circuit distinct from token-budget regulation.

**Head interventions enable causal steering** (Exp. 3): Scaling length heads modifies generation length by $\pm 30\%$ with minimal accuracy loss. Depth-controlled reasoning-head scaling yields a perception-reasoning trade-off: low-layer amplification improves perception-dominant examples (+4.1%), high-layer amplification improves reasoning-dominant examples (+5.3%). These effects generalize across five benchmarks (MMMU, MathVista, ScienceQA, ChartQA, A-OKVQA).

**Contributions** (1) We formalize *reasoning helpfulness* and *desired length* as practical control targets for VLM inference and show they are linearly decodable from last-token embeddings. (2) We localize these signals to partially disentangled attention-head circuits: length control in mid-layers, reasoning/difficulty assessment in higher layers. (3) We demonstrate causal control via head-scaling interventions, enabling test-time steering of generation length and the perception-reasoning balance without retraining.

## 2. Related Work

### 2.1. Visual Reasoning

A growing line of work investigates how to make multimodal reasoning more explicit, grounded, and controllable. Multimodal-CoT (Zhang et al., 2023) separates rationale generation from answer prediction to reduce hallucination, while tool- and program-augmented agents (VisProg (Gupta & Kembhavi, 2023), ViperGPT (Surís et al., 2023)) offload intermediate computation to external modules for improved faithfulness. More interactive "reason-and-act" formulations extend ReAct-style prompting to multimodal settings (MM-ReAct (Yang et al., 2023)). Recent efforts also aim to move beyond purely textual rationales by incorporating visual state transitions (Uni-CoT (Qin et al., 2026)) or "visualization of thought" for spatial reasoning (MVoT (Li et al., 2025)), and by reducing supervision via preference-based visual CoT learning (UV-CoT (Zhao et al., 2025)). Overall, these trends motivate studying visual reasoning not only as answering isolated questions, but as sustained, grounded, multi-step inference under realistic constraints.

### 2.2. Effective Test-Time Scaling in Reasoning Models

Test-time scaling improves models by allocating more inference-time compute, such as generating longer rationales or exploring multiple solutions. Chain-of-Thought prompting (Wei et al., 2022) and its extensions (e.g., self-consistency (Wang et al., 2023), least-to-most (Zhou et al., 2023)) enable stepwise reasoning, while search-based methods (Tree-of-Thoughts (Yao et al., 2023), Forest-of-

Thought (Bi et al., 2024), graph-based approaches (Besta et al., 2024)) explicitly explore reasoning paths. Selection mechanisms, such as process supervision, verifiers (Lightman et al., 2024), and adaptive compute allocation (Snell et al., 2025), help choose among multiple traces. Large Reasoning Models such as o1 (OpenAI, 2024a;b) and DeepSeek-R1 (Guo et al., 2025) leverage RL and long-form reasoning, with benchmarks (GSM8K (Cobbe et al., 2021), LiveCodeBench (Jain et al., 2025)) driving progress.

However, longer reasoning is not always beneficial: excessive length can degrade accuracy through overthinking (Su et al., 2025a; Cuadron et al., 2025). Methods such as TOPS (Yang et al., 2025) and LCPO (Aggarwal & Welleck, 2025) optimize rationale length, while REST (Pan et al., 2025b) and recent surveys (Sui et al., 2025) highlight challenges in efficient reasoning. Alternatives to longer CoT include adaptive/parallel programs (Pan et al., 2025a), agent-based scaling (Zhu et al., 2025b), test-time parameter updates (Akyürek et al., 2025), and precomputation (Lin et al., 2025). R-Horizon (Lu et al., 2026) probes long-horizon reasoning and informs RL-based improvements. Overall, test-time scaling is fundamentally an allocation problem: determining how to allocate limited compute for robust, grounded, multi-step reasoning.

## 3. Method

### 3.1. Problem Setup

We investigate whether a vision-language model (VLM) can internally represent *when thinking helps* and *how much thinking is desired* for a given multimodal question. Each data sample is a pair $(I, x)$, where $I$ is the image (or multi-image context) and $x$ is the question text with multiple-choice options and a gold answer $a^\star$.

To systematically analyze the model, for each base model family and size, we evaluate two aligned variants: (i) an *Instruct* variant that answers directly, and (ii) a *Reasoning/Thinking* variant that is tuned to produce explicit reasoning before the final answer. We denote these modes as $m \in \{\text{INST}, \text{THINK}\}$.

For every example $(I, x)$ and mode $m$, we run $K = 10$ stochastic generations to estimate correctness as a probability rather than from a single sample outcome. Let the $k$-th decoded output under mode $m$ be $y_m^{(k)}$, and let $\text{Ans}(y)$ extract the final multiple-choice option (e.g., A/B/C/D) from $y$. We define the pass-level correctness indicator $c_m^{(k)} = \mathbb{K}[\text{Ans}(y_m^{(k)}) = a^\star]$. The empirical correctness rate for mode $m$ is $\hat{p}_m = \frac{1}{K} \sum_{k=1}^{K} c_m^{(k)}$.

**Reason helpfulness.** We measure whether thinking helps on an instance by comparing correctness rates: $r_{\text{help}} = \hat{p}_{\text{THINK}} - \hat{p}_{\text{INST}}$, $r_{\text{help}} \in [-1, 1]$. Positive values indicate

| $K$ | corr($r_K, r_{40}$) ↑ | MAE($\ell_K, \ell_{40}$) ↓ |
|---|---|---|
| 5 | 0.885 | 63 |
| 7 | 0.935 | 38 |
| 10 | 0.962 | 25 |
| 20 | 0.984 | 10 |
| 30 | 0.971 | 5 |
| 40 | 0.985 | 2 |

*Table 1.* Sensitivity of the empirical labels to the number of stochastic passes on Qwen3-VL-4B, using $K = 40$ as the reference. Here $\ell_K = \ell_{\text{opt},K}$. The targets stabilize quickly, with most of the gain obtained by $K = 7$–10.

that the Thinking variant is more reliable on this example.

**Desired length.** We define the *desired length* as the *minimum* length among all correct generations produced by the *winning* mode for that example. Let $\ell_m^{(k)}$ be the generated length (in tokens) of $y_m^{(k)}$. For each mode $m$, define the set of lengths of correct generations as $\mathcal{L}_m = \{\ell_m^{(k)} : c_m^{(k)} = 1\}$. If $\mathcal{L}_m \neq \emptyset$, define the minimum correct length as $\ell_m^{\min} = \min(\mathcal{L}_m)$; otherwise, set $\ell_m^{\min} = 0$.

We select the winner mode $m^\star$ by correctness, breaking ties by choosing the mode with the *smaller minimum correct length*:

$$m^\star = \begin{cases} \text{THINK} & \text{if } \hat{p}_{\text{THINK}} > \hat{p}_{\text{INST}}, \\ \text{INST} & \text{if } \hat{p}_{\text{THINK}} < \hat{p}_{\text{INST}}, \\ \arg\min_{m \in \{\text{INST}, \text{THINK}\}} \ell_m^{\min} & \text{if } \hat{p}_{\text{THINK}} = \hat{p}_{\text{INST}}. \end{cases}$$

Thus, when both modes are equally correct, we select the mode that achieves correctness with fewer tokens. The *desired length label* is then $\ell_{\text{opt}} = \ell_{m^\star}^{\min}$.

Each sample additionally includes a pre-existing difficulty label $d$ from the dataset annotation, rather than from our generations. In the probe experiments below, $d$ is treated as an auxiliary target for analysis and control.

**Scope of the oracle labels.** The targets above are empirical control labels, not intrinsic properties of a question. In particular, $r_{\text{help}}$ is defined relative to a fixed model family, decoding protocol, and pair of variants, while $\ell_{\text{opt}}$ is the shortest observed correct generation among $K$ samples rather than a globally minimal proof length. This distinction matters because improved answer accuracy does not by itself imply improved visual grounding: a Thinking response can still benefit from language priors, or fail when its rationale drifts away from the image. We therefore interpret the probes as signals for adaptive inference under a controlled protocol, and evaluate interventions on held-out examples rather than treating the labels as universal difficulty annotations.

As a sanity check on the finite-sampling labels, Table 1 compares estimates from smaller $K$ to a $K = 40$ reference

under fixed decoding. We keep temperature and top-$p$ fixed in this analysis so that changes in the table reflect the number of sampled completions rather than a different generation regime. The high correlation for $r_{\text{help}}$ and declining MAE for $\ell_{\text{opt}}$ support $K = 10$ as a practical compromise between label stability and evaluation cost.

## 3.2. Probes from Last-Token Representations

Prior work has demonstrated that the last-token hidden representation can encode task-relevant properties that are linearly recoverable, even before generating an answer. Motivated by this finding, we probe our selected models using the same approach.

Given a multimodal prompt constructed from $(I, x)$ using the model's chat template, we run the model forward in prefill and record the final-layer hidden state of the last prompt token. Denote this vector by $h \in \mathbb{R}^D$. We extract $h_{\text{INST}}$ from the Instruct variant and $h_{\text{THINK}}$ from the Thinking variant. This probe interface is intentionally conservative: it gives a standardized pre-generation state for comparing Instruct and Thinking variants under identical supervision. It does not imply that earlier layers, image tokens, or states produced during the reasoning trace are unimportant; rather, it tests whether the decision-relevant signal is already available before the model commits to a long answer.

Rather than using a multi-task head, we train **three independent probes** for better interpretability, each optimized for a single target:

$$t \in \{r_{\text{help}}, \ell_{\text{opt}}, d\}. \tag{1}$$

For $r_{\text{help}}$ and $\ell_{\text{opt}}$, we use linear regression:

$$\hat{y}_t = w_t^\top h + b_t, \qquad \mathcal{L}_t = \frac{1}{N} \sum_{i=1}^{N} \left(y_{t,i} - \hat{y}_{t,i}\right)^2. \tag{2}$$

Difficulty is a categorical label $d \in \{\text{EASY}, \text{MEDIUM}, \text{HARD}\}$. We therefore train a **separate linear classifier**:

$$p(d \mid h) = \text{softmax}(Wh + b), \quad \mathcal{L}_d = -\log p(d \mid h).$$

For each target $t$, we train **two versions** of the probe that differ only in the representation used. Both versions predict the *same labels* $y_t$ computed from the multi-pass procedure (Section 3.1); thus, any performance gap reflects differences in linear decodability between Instruct and Thinking representations.

**How the probes are used.** At test time, the probe requires only a single prefill pass over the image-question prompt. This makes it usable before committing to a long Thinking generation. A simple controller can first predict $\hat{r}_{\text{help}}$ and $\hat{\ell}_{\text{opt}}$, then choose among three actions: answer directly, generate a Thinking response with an adjusted token budget, or leave the model unchanged when the probe confidence is low. In our experiments, the probes are used primarily as analysis tools and as sources for head attribution; however, this inference-time view is important because it separates the cost of deciding *whether* to reason from the much larger cost of generating a full rationale.

**Avoiding label leakage.** All probe training, head ranking, and intervention selection are performed using only the training split or validation split. Held-out test examples are used only for reporting probe accuracy, head-overlap statistics, and intervention effects. Thus, the causal steering experiments do not select heads or scaling strengths based on the examples on which the final gains are reported.

# 4. Experiment 1: Thinking Models Encode Meta-Cognitive Signals More Linearly

## 4.1. Motivation

Having established our multi-pass evaluation procedure and probe targets (Section 3), we now ask: do reasoning-tuned models organize their representations in a way that makes these meta-cognitive properties, namely *when thinking helps* ($r_{\text{help}}$) and *how much to generate* ($\ell_{\text{opt}}$), more linearly recoverable? We probe the final-layer last-token embedding from both Instruct and Thinking variants, training separate linear probes on identical labels derived from the multi-pass procedure. Any performance gap thus reflects differences in *linear decodability*, not differences in ground-truth labels.

## 4.2. Setup

We use the MMMU validation set (`MMMU-val900`) as our base dataset due to its popularity and diverse multimodal categories (Yue et al., 2024). Each example provides an image-question pair $(I, x)$ with ground-truth answer $a^\star$ and a difficulty label $d$ from our annotation pipeline, as described in Section 3.1.

## 4.3. Model Selection

Recent open-source VLMs often generate step-by-step rationales, but improved rationale format does not guarantee better reasoning (Zhang et al., 2025; Wang et al., 2025a). Prior work highlights that models may imitate successful reasoning traces without truly recognizing or avoiding errors, and that visual rationales can create an illusion of thinking without genuine perceptual gains. RL-based post-training can help models discover more effective reasoning strategies (Su et al., 2025b).

Given these concerns, we select two VLM families, InternVL and Qwen3-VL, that (i) report strong multimodal

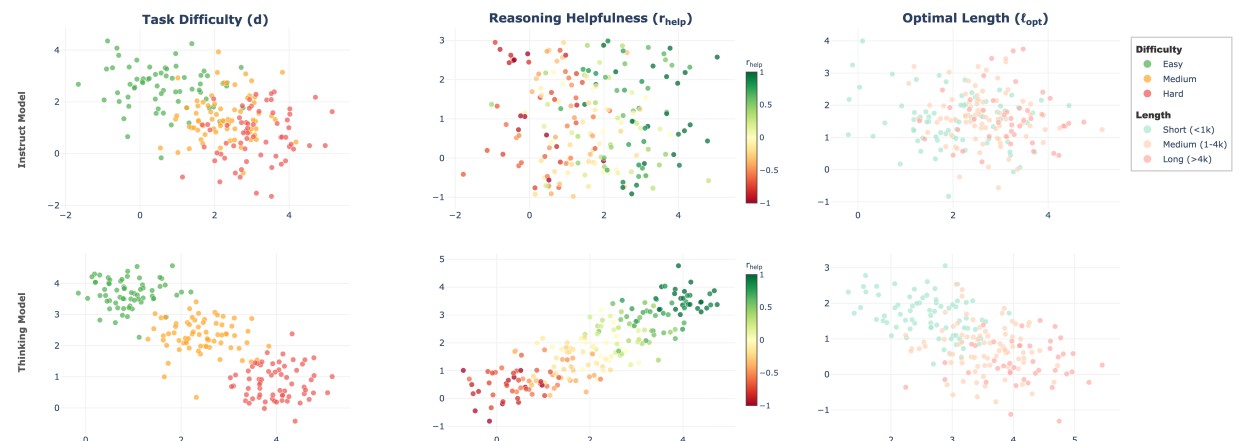

*Figure 2.* **Experiment 1: Probing meta-cognitive signals from the final embedding. (Left)** Task difficulty clustering measured as embedding separation across easy, medium, and hard questions. Thinking models exhibit consistently larger separation than Instruct models, indicating stronger difficulty-aware representations. **(Middle)** Predicted reasoning helpfulness ($r_{\text{help}}$) versus ground truth. Probes trained on Thinking model embeddings align more closely with the diagonal, suggesting improved encoding of when additional reasoning is beneficial. **(Right)** Predicted desired response length ($\ell_{\text{opt}}$) versus ground truth (log scale). Both models capture length variation, but Thinking models show tighter alignment with the oracle line, indicating more accurate internal length control signals.

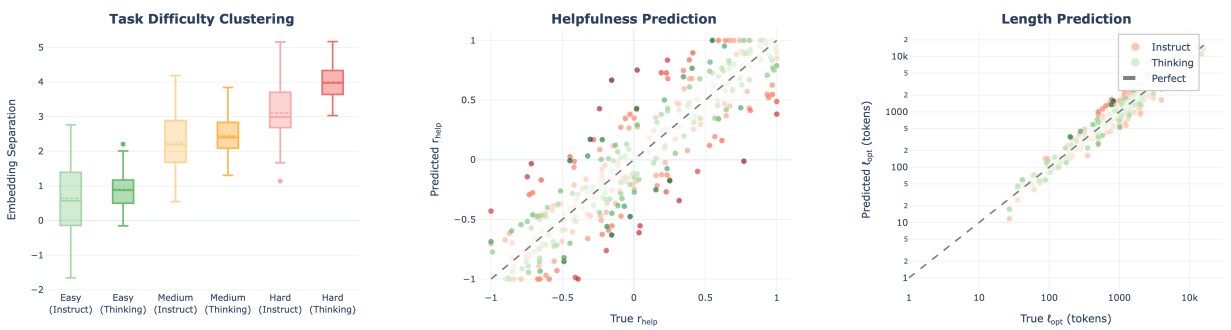

*Figure 3.* **Experiment 1: Representation geometry of Instruct vs. Thinking models.** We visualize the final-token embedding space for both models under three probing targets. **(Left)** Task difficulty ($d$): Thinking models exhibit clearer stratification between easy, medium, and hard questions than Instruct models. **(Middle)** Reasoning helpfulness ($r_{\text{help}}$): embeddings from Thinking models form a smooth gradient aligned with $r_{\text{help}}$, whereas Instruct embeddings show weaker structure. **(Right)** Desired length ($\ell_{\text{opt}}$): Thinking models better organize examples by target response length, while Instruct models show greater overlap across length regimes. Overall, Thinking models encode difficulty, reasoning utility, and length in a more disentangled and linearly decodable manner.

reasoning on MMMU (Yue et al., 2024; Zhu et al., 2025a; Bai et al., 2025), and (ii) introduce explicit reasoning-oriented training (e.g., Cascade RL in InternVL3.5 (Wang et al., 2025b), architectural/training upgrades in Qwen3-VL (Bai et al., 2025)). Both provide INSTRUCT and THINKING variants, enabling controlled comparisons.

Our primary grid evaluates InternVL-{4B, 8B} and Qwen3-VL-{4B, 8B}, each with both variants (8 total), balancing accessibility for probing with sufficient scale for reasoning-tuned representations (Lee et al., 2025). We additionally include Qwen3-VL-32B as a larger-scale check within the Qwen family; InternVL does not provide a dense 32B counterpart in this setting. The evaluated models are InternVL-4B, InternVL-8B, Qwen3-VL-4B, Qwen3-VL-8B, and Qwen3-VL-32B, each with Instruct and Thinking variants.

**Generation and labeling.** For each model variant and each MMMU example, we run $K = 10$ passes and compute ($r_{\text{help}}, \ell_{\text{opt}}$) as described in Section 3.1. We maintain consistent decoding settings across variants (sampling enabled with fixed temperature and top-$p$ parameters).

**Probe training protocol.** For each *base family/size* (e.g., InternVL-4B), we train two versions of each probe: 1. an *Instruct-probe* trained on $h_{\text{INST}}$, 2. a *Thinking-probe* trained on $h_{\text{THINK}}$, where both predict the *same ground-truth labels* ($r_{\text{help}}, \ell_{\text{opt}}, d$) computed via our multi-pass procedure. We use a 70/15/15 train/validation/test split of `MMMU-val900`. We report: 1. for $r_{\text{help}}$: MSE and Spearman correlation ($\rho$), 2. for $\ell_{\text{opt}}$: MSE and MAE, 3. for $d$: classification accuracy.

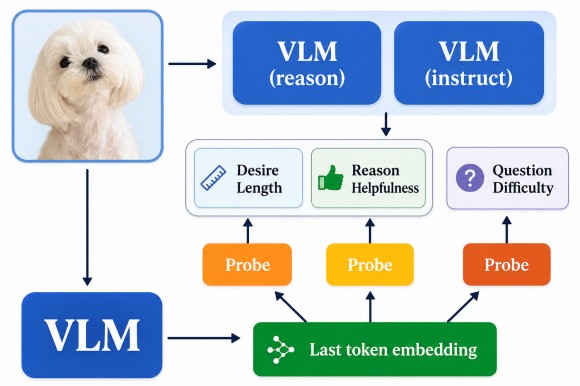

*Figure 4.* **Experiment 1 (Probing setup).** Given an image-question input, we run two VLM variants (Reason/Thinking vs. Instruct) and extract the final *last-token embedding* before decoding. Linear probes are trained on this embedding to predict three meta signals: *desired response length* ($\ell_{opt}$), *reasoning helpfulness* ($r_{help}$; whether additional reasoning is beneficial), and *question difficulty* ($d$). This enables a controlled comparison of what each model encodes about length control and when reasoning is needed without changing the generation procedure.

### 4.4. Results

Across the tested model families and sizes, probes trained on *Thinking* representations are consistently more accurate than probes trained on *Instruct* representations for *both* targets: 1. *Lower error:* Thinking-probes achieve lower MSE/MAE for $r_{help}$ and $\ell_{opt}$. 2. *Higher rank consistency:* Thinking-probes achieve higher Spearman $\rho$ on $r_{help}$, indicating better instance-level ordering of "thinking helps vs. hurts". 3. *Scaling trend:* The gap is larger at 8B than 4B and persists in Qwen3-VL-32B, suggesting that increased capacity and reasoning-oriented tuning make these signals more linearly recoverable within the tested families.

### 4.5. Interpretation

A plausible interpretation is that reasoning-oriented training not only improves final-answer quality but also reorganizes internal representations so that meta-cognitive properties, such as whether additional reasoning is beneficial ($r_{help}$) and the appropriate compute budget ($\ell_{opt}$), become more explicitly encoded and thus more linearly decodable from the final-layer last-token embedding. This aligns with prior findings that complex behaviors may appear non-separable in low-dimensional visualizations yet become linearly recoverable in the model's native high-dimensional representation space, and that such linear structure emerges more strongly in well-trained or reasoning-tuned models.

The 32B row should not be read as evidence of universality across all VLMs, but it addresses the most direct scale concern within our available model families: relative to Qwen3-VL-8B, Qwen3-VL-32B improves all eight probe

metrics, with the largest gain in Thinking-mode prediction of $r_{help}$. At the same time, the absolute Spearman correlations remain moderate, so the claim is comparative rather than that a final-token linear probe perfectly recovers reasoning utility. This is expected because $r_{help}$ is a difference between two empirical correctness rates and is estimated from finite stochastic samples.

Having established that Thinking models encode these signals more linearly, we next investigate *where* in the network these signals are computed (Experiment 2, Section 5) and whether we can *causally steer* them via targeted head interventions (Experiment 3, Section 6).

## 5. Experiment 2: Heads Related to Probes

### 5.1. Motivation

Experiment 1 (Section 4) demonstrates that THINKING variants encode meta-cognitive signals, whether reasoning helps ($r_{help}$), desired generation length ($\ell_{opt}$), and difficulty ($d$), in a more linearly decodable manner than INSTRUCT variants. This improved encoding suggests that reasoning-tuned models develop specialized internal circuitry for these computations.

We now investigate: Which attention heads causally support these meta-cognitive signals? Do different signals (length control vs. reasoning/difficulty assessment) rely on distinct or overlapping circuits? Our analysis is inspired by recent work showing that difficulty perception can be localized to a small set of heads and causally modified via head-wise scaling interventions (Lee et al., 2025).

### 5.2. Setup

**Models and data.** We use the same eight VLM variants as in Experiment 1: InternVL-{4B,8B} and Qwen3-VL-{4B,8B}, each with INSTRUCT and THINKING variants, evaluated on `MMMU-val900`.

**Probe targets.** We reuse the three targets from Section 3.1: reasoning helpfulness $r_{help}$, desired length $\ell_{opt}$, and difficulty $d$. Following Section 3.2, we train separate linear probes (regression for $r_{help}$ and $\ell_{opt}$, classification for $d$) on the final-layer last-token embedding $h_{THINK}$ from each Thinking variant. **Perception-vs-reasoning cohorts.** To quantify the "perception vs. reasoning" trade-off, we form two cohorts: **Perception-dominant** samples: bottom-$q$ quantile of $r_{help}$ (extra reasoning is unhelpful / hurts). **Reasoning-dominant** samples: top-$q$ quantile of $r_{help}$. In practice we use $q \in \{0.2, 0.3\}$ and report robustness.

### 5.3. Head-wise Attribution Across Layers

We assign each head $(\ell, i)$ an attribution score for each probe target by measuring how the probe prediction changes when

| Base model | $r_{\text{help}}$ | | | | $\ell_{\text{opt}}$ | | | |
| | Instruct-probe | | Thinking-probe | | Instruct-probe | | Thinking-probe | |
| | MSE ↓ | $\rho$ ↑ | MSE ↓ | $\rho$ ↑ | MSE ↓ | MAE ↓ | MSE ↓ | MAE ↓ |
|---|---|---|---|---|---|---|---|---|
| InternVL-4B | 0.182 | 0.324 | **0.141** | **0.412** | 3247 | 42.3 | **2856** | **38.7** |
| InternVL-8B | 0.168 | 0.341 | **0.119** | **0.458** | 3108 | 40.1 | **2531** | **35.2** |
| Qwen3-VL-4B | 0.194 | 0.298 | **0.153** | **0.389** | 3512 | 45.8 | **3089** | **41.2** |
| Qwen3-VL-8B | 0.173 | 0.328 | **0.126** | **0.447** | 3294 | 42.9 | **2647** | **36.8** |
| Qwen3-VL-32B | 0.157 | 0.362 | **0.093** | **0.531** | 3012 | 39.1 | **2142** | **31.8** |

*Table 2.* Probe performance on MMMU. Thinking-probes consistently achieve lower MSE/MAE and higher Spearman correlation across models. The Qwen3-VL-32B result provides an additional scale check and shows the strongest Thinking-mode decodability in this table.

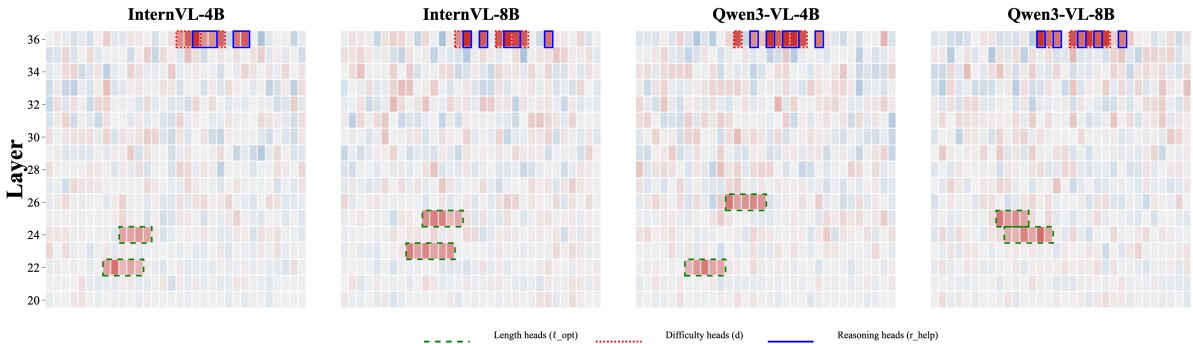

*Figure 5.* Head-wise attribution maps for length, difficulty, and reasoning across models. Head attribution scores over *layer* (y-axis) and *attention head* (x-axis) for InternVL-4B, InternVL-8B, Qwen3-VL-4B, and Qwen3-VL-8B, computed as in Experiment 2. Highlighted regions indicate the top-$K$ heads for each attribute: length-control heads ($\ell_{\text{opt}}$, green dashed), difficulty heads ($d$, red dotted), and reasoning-helpfulness heads ($r_{\text{help}}$, blue solid). Across all models, length heads consistently concentrate in middle-to-lower layers, while reasoning and difficulty heads cluster in higher layers with substantial overlap, suggesting a shared circuit for difficulty perception and reasoning control, distinct from token-budget regulation.

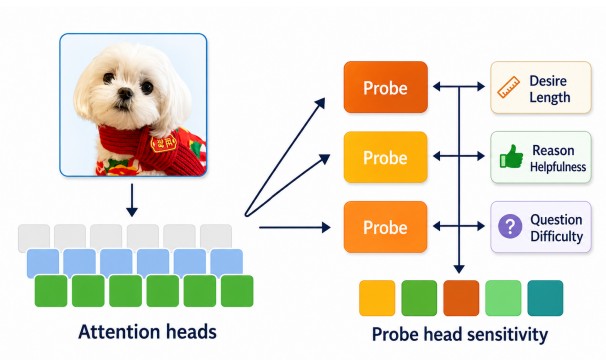

*Figure 6.* **Experiment 2 (Head attribution / probe sensitivity).** To localize the circuit supporting each probe signal, we compute *probe head sensitivity* by intervening on individual attention heads (e.g., scaling/ablation) and measuring the resulting change in the probe prediction. The sensitivity map ranks heads by how causally they affect predicted *desired length*, *reasoning helpfulness*, and *difficulty*. We find that length-related heads tend to concentrate in middle-to-lower layers, while heads associated with reasoning helpfulness and difficulty substantially overlap, suggesting a shared mechanism that couples perceived difficulty with the decision to engage in deeper reasoning.

that head is ablated or isolated.

**Ablation-based attribution.** Let $\hat{y}(\cdot)$ be a trained probe (for $d$, $r_{\text{help}}$, or $\ell_{\text{opt}}$), and let $h$ be the final-layer last-token embedding produced by the unmodified model. For each layer $\ell$ and head $i$, we intervene by scaling the head output in the multi-head attention tensor before the output projection: $H^{(\ell)}_{:,:,i,:} \leftarrow \alpha \cdot H^{(\ell)}_{:,:,i,:}$ with $\alpha = 0$ for ablation and $\alpha = 1$ otherwise, producing a modified final embedding $h^{(-\ell,i)}$. We define the head's **causal contribution** to the probe as: $A_{\ell,i} = \mathbb{E}_b\left[\hat{y}(h_b) - \hat{y}(h_b^{(-\ell,i)})\right]$. For cohort-style contrasts (e.g., easy vs. hard, perception- vs. reasoning-dominant), we compute $\Delta_{\ell,i} = A_{\ell,i}^{(\text{group1})} - A_{\ell,i}^{(\text{group2})}$.

**Ranking and selection.** For each target, we rank heads by $|\Delta_{\ell,i}|$ and take the top-$K$ heads ($K \in \{8, 16\}$) as the *control set* for that attribute.

### 5.4. Length vs. Reasoning/Difficulty Heads

**Length-control heads concentrate in middle-to-lower layers.** Across both model families and variants, the highest-attribution heads for $\ell_{\text{opt}}$ consistently appear in the **middle-to-lower** transformer layers, rather than in the earliest layers. This suggests that token-budget control is implemented after initial perceptual parsing but before final decoding decisions.

**Reasoning-helpfulness and difficulty heads overlap substantially.** In contrast, the head sets with highest attribution

| $K$ | $J(S^\ell, S^r)$ | $J(S^r, S^d)$ | $\bar{L}(S^\ell)$ | $\bar{L}(S^r)$ |
|---|---|---|---|---|
| 8 | $0.11_{\pm 0.03}$ | $0.71_{\pm 0.04}$ | $15.2_{\pm 2.1}$ | $26.8_{\pm 1.8}$ |
| 12 | $0.13_{\pm 0.04}$ | $0.69_{\pm 0.03}$ | $14.8_{\pm 2.3}$ | $27.1_{\pm 1.5}$ |
| 16 | $0.12_{\pm 0.03}$ | $0.72_{\pm 0.03}$ | $15.5_{\pm 2.0}$ | $26.5_{\pm 1.6}$ |
| 20 | $0.14_{\pm 0.04}$ | $0.68_{\pm 0.04}$ | $14.9_{\pm 2.2}$ | $27.3_{\pm 1.7}$ |

*Table 3.* Robustness to $K$: Jaccard overlap $J$ and mean layer index $\bar{L}$ of the length ($S^\ell$), reasoning ($S^r$), and difficulty ($S^d$) head sets, averaged across models. Patterns stay stable across head-set sizes.

| Model | $\mathbf{J}(S^\ell, S^r)$ | $\mathbf{J}(S^r, S^d)$ | $\mathbf{J}(S^\ell, S^d)$ |
|---|---|---|---|
| IVL-4B-T | 0.12 | **0.68** | 0.19 |
| IVL-8B-T | 0.09 | **0.73** | 0.14 |
| QV3-4B-T | 0.15 | **0.71** | 0.21 |
| QV3-8B-T | 0.11 | **0.76** | 0.17 |

*Table 4.* Jaccard overlap between top-16 head sets. $S^\ell$, $S^r$, $S^d$ denote length, reasoning, and difficulty head sets. Reasoning and difficulty heads show high overlap (0.68–0.76), while length heads are largely disjoint.

for $r_{\text{help}}$ and difficulty $d$ exhibit substantial overlap (high Jaccard similarity in the top-$K$ sets), indicating that the computations underlying reasoning assessment and difficulty perception are supported by closely shared circuitry. This aligns with the hypothesis that difficulty perception acts as an internal trigger for adaptive reasoning.

### 5.5. Ablation: Robustness to Head Selection Size $K$

To verify that our findings are not artifacts of a particular choice of $K$, we repeat the analysis with $K \in \{8, 12, 16, 20\}$. Table 3 shows that the Jaccard overlap between reasoning and difficulty heads remains consistently high ($> 0.65$) across all values of $K$, while length heads remain largely disjoint (Jaccard $< 0.25$) from both reasoning and difficulty heads.

**Summary and next step.** We have identified two partially disentangled circuits: *length-control heads* in middle-to-lower layers and *reasoning/difficulty heads* in higher layers with substantial overlap. Having localized these circuits via attribution analysis, Experiment 3 (Section 6) tests whether they are *causally* responsible by directly intervening on them during generation.

## 6. Experiment 3: Causal Reasoning Steering

### 6.1. Motivation

Experiment 2 (Section 5) localizes two partially disentangled circuits via probe-based attribution:

1. **Length-control heads** ($S^{\text{len}}$): most predictive of desired generation length ($\ell_{\text{opt}}$), concentrated in *middle-to-lower* layers.

2. **Reasoning/difficulty heads** ($S^{\text{reas}}$): most predictive of reasoning helpfulness ($r_{\text{help}}$) and difficulty ($d$), with substantial overlap between the two signals, concentrated in *higher* layers.

Attribution analysis reveals *correlation* between head activations and probe predictions but does not establish *causality*. We now test whether these circuits are causally responsible for length control and reasoning decisions by directly intervening on them during generation, following head-wise scaling methods used to manipulate perceived difficulty in LLMs (Lee et al., 2025).

### 6.2. Head-wise Scaling Intervention

Let $H^{(\ell)} \in \mathbb{R}^{B \times L \times N \times d}$ denote the multi-head attention output at layer $\ell$ (batch $B$, sequence length $L$, heads $N$, head dim $d$). We intervene by multiplying selected heads in $H^{(\ell)}$ *before* the output projection, i.e., the same operational point as prior work (Lee et al., 2025):

$$H^{(\ell)}_{:,:,i,:} \leftarrow \begin{cases} \alpha_\downarrow \, H^{(\ell)}_{:,:,i,:} & i \in S_{\text{down}} \\ \alpha_\uparrow \, H^{(\ell)}_{:,:,i,:} & i \in S_{\text{up}} \\ H^{(\ell)}_{:,:,i,:} & \text{otherwise} \end{cases} \quad (3)$$

**Which head sets do we scale?** All head sets are derived from the attribution ranking in Experiment 2 (Section 5.3), computed on the training split:

**Length heads** $S^{\text{len}}$: top-$K$ heads by $|A_{\ell,i}|$ for the $\ell_{\text{opt}}$ probe (Eq. 5.3).

**Reasoning heads** $S^{\text{reas}}$: top-$K$ heads by $|A_{\ell,i}|$ for the $r_{\text{help}}$ probe. Due to the strong overlap with difficulty-attributed heads observed in Experiment 2, we treat these as a shared reasoning/difficulty circuit.

**Perception heads** $S^{\text{perc}}$: top-$K$ heads with *opposite-sign* attribution on the perception-dominant vs. reasoning-dominant cohort contrast. Concretely, for each head we compute $\Delta^r_{\ell,i} = A^{\text{perc}}_{\ell,i} - A^{\text{reas}}_{\ell,i}$ for the $r_{\text{help}}$ probe, and select heads in the direction opposite to $S^{\text{reas}}$. This set is not intended to name a separate semantic module; it is a signed contrast set used to test whether pushing away from reasoning-dominant heads recovers more perception-centered behavior.

We use $K = 16$ for all experiments unless otherwise stated.

**Low-vs-high layer reasoning subsets.** To test the depth effect, we split reasoning heads by layer index: $S^{\text{reas}}_{\text{low}}$ (lower layers) and $S^{\text{reas}}_{\text{high}}$ (upper layers). We apply Eq. 3 either to the low subset or the high subset, keeping the other subset unchanged. Operationally, heads in layers with index smaller than $\lfloor L_{\text{layers}}/2 \rfloor$ are assigned to the low group, and the remaining selected reasoning heads are assigned to the high group.

**Intervention conditions.** We use three families of interventions: 1. **Length knob:** $S_{\text{up}} = S^{\text{len}}$, $S_{\text{down}} = \emptyset$ with $\alpha_\uparrow \in \{0.5, 1.0, 1.5, 2.0\}$ (and optionally symmetric suppression with $\alpha_\downarrow < 1$). 2. **Reasoning in-**

| Model | 0.5 | 1.0 | 1.5 | 2.0 | **Acc change** |
|---|---|---|---|---|---|
| | *Mean generation length (tokens)* | | | | |
| IVL-4B-T | 127 | 218 | 342 | 489 | $-1.2\%$ |
| IVL-8B-T | 139 | 235 | 368 | 521 | $-0.8\%$ |
| QV3-4B-T | 115 | 203 | 318 | 456 | $-1.5\%$ |
| QV3-8B-T | 131 | 227 | 351 | 498 | $-0.9\%$ |

*Table 5.* Length control via $S^{\text{len}}$ scaling. Baseline ($\alpha = 1.0$) is unmodified model. Accuracy change is measured between $\alpha = 0.5$ and $\alpha = 2.0$, showing length control has minimal impact on correctness within this range.

| Model | Perc. | Reas. | Overall | change |
|---|---|---|---|---|
| | $\Delta$Acc (low) | $\Delta$Acc (high) | $\Delta$Acc | |
| IVL-4B-T | $+3.2\%$ | $+4.1\%$ | $+0.8\%$ | $+28\%$ |
| IVL-8B-T | $+4.1\%$ | $+5.3\%$ | $+1.2\%$ | $+31\%$ |
| QV3-4B-T | $+2.8\%$ | $+3.7\%$ | $+0.6\%$ | $+25\%$ |
| QV3-8B-T | $+3.6\%$ | $+4.9\%$ | $+1.1\%$ | $+29\%$ |

*Table 6.* Depth-controlled steering summary. "low" = scaling $S_{\text{low}}^{\text{reas}}$ benefits perception-dominant examples; "high" = scaling $S_{\text{high}}^{\text{reas}}$ benefits reasoning-dominant examples. Length change measures increase vs. baseline.

**crease / perception increase:** (1)*Reasoning-increase:* $S_{\text{up}} = S^{\text{reas}}$, $S_{\text{down}} = S^{\text{perc}}$. (2)*Perception-increase:* $S_{\text{up}} = S^{\text{perc}}$, $S_{\text{down}} = S^{\text{reas}}$. 3. **Depth-controlled reasoning steering:** same as above, but with $S_{\text{low}}^{\text{reas}}$ vs. $S_{\text{high}}^{\text{reas}}$ used as $S_{\text{up}}$. The push–pull formulation is important because increasing a reasoning-related set alone can mostly increase verbosity; the signed contrast tests steering along the perception–reasoning axis. The reasoning-increase and perception-increase settings verify the intervention direction, while the depth-controlled setting is the main condition used to separate perception-dominant from reasoning-dominant behavior in the result tables. Unless otherwise stated, we set $(\alpha_\uparrow, \alpha_\downarrow) = (2.0, 0.1)$ to match the strong-causal setting used in prior head-scaling work (Lee et al., 2025), and we also report milder settings for stability.

### 6.3. Metrics

We evaluate the effects of head-scaling interventions along three dimensions: **Generation length** (in tokens) and alignment with oracle desired length $\ell_{\text{opt}}$ (mean absolute difference). **Answer accuracy** on MMMU and additional perception+reasoning benchmarks (MathVista, ScienceQA, ChartQA, A-OKVQA). **Cohort-specific accuracy:** performance on perception-dominant vs. reasoning-dominant subsets, defined by the top/bottom $q = 0.2$ quantiles of $r_{\text{help}}$ as in Section 5.

**Calibration and deployment interpretation.** Strong scales test causality, while milder scales are more appropriate for controlled generation. In deployment, thresholds, token budgets, and scaling coefficients should be calibrated on held-out data, with fallback to the unmodified model when probe margins are small. Thus, the gains are lightweight causal evidence under matched conditions, not universal heads or coefficients for arbitrary VLMs.

**Results I: Length Control** Scaling length heads produces a monotonic change in output length: increasing $\alpha_\uparrow$ for $S^{\text{len}}$ reliably increases generated tokens, while suppressing $S^{\text{len}}$ shortens responses. Notably, within a moderate range of scales (e.g., 0.5–1.5), accuracy remains relatively stable, suggesting that the length circuit is partially separable from correctness.

**Results II: Depth-Dependent Reasoning Steering** We ob-

serve a consistent **depth-dependent steering** phenomenon:

*Emphasizing reasoning heads in lower layers increases perceptual grounding.* When we apply the reasoning-increase intervention to $S_{\text{low}}^{\text{reas}}$, performance improves on the perception-dominant cohort (where additional reasoning tends to be detrimental), and qualitative outputs become more visually grounded and less prone to overthinking.

*Emphasizing reasoning heads in higher layers increases abstract reasoning.* Conversely, scaling $S_{\text{high}}^{\text{reas}}$ improves performance on the reasoning-dominant cohort, and outputs exhibit more multi-step symbolic manipulation and longer dependency chains.

*Reasoning and difficulty heads behave as a shared circuit.* Because the top heads for $r_{\text{help}}$ and $d$ overlap strongly (Exp2), we find similar steering effects when selecting heads via either probe as both choices yield mode shifts in Eq. 3.

**Results III: Cross-Dataset Generalization** We repeat the same intervention protocol on benchmarks that tightly couple visual perception with multi-step reasoning: MathVista, ScienceQA (Lu et al., 2022), ChartQA (Masry et al., 2022), and A-OKVQA. Across datasets, we observe the same qualitative pattern: length heads control token budget, while reasoning-head scaling exhibits a depth-dependent shift between perceptual grounding and abstract reasoning.

| Dataset | Baseline Acc | Length $\Delta$Len | low $\Delta$Acc | high $\Delta$Acc |
|---|---|---|---|---|
| MMMU | 62.4% | +31% | +4.1% | +5.3% |
| MathVista | 54.7% | +28% | +3.8% | +4.9% |
| ScienceQA | 78.3% | +25% | +2.9% | +3.7% |
| ChartQA | 69.1% | +33% | +3.5% | +4.6% |
| A-OKVQA | 71.2% | +29% | +3.2% | +4.2% |

*Table 7.* Cross-dataset generalization (InternVL-8B-T). All benchmarks exhibit the same pattern: length heads control generation length, low-layer reasoning heads benefit perception-dominant examples, high-layer reasoning heads benefit reasoning examples.

## Conclusion

We demonstrated that vision-language models internally signal when reasoning is needed and how much computation to use. Mechanistic analysis identified attention-head circuits for length control and reasoning, enabling targeted, causal interventions that steer model behavior without retraining.

## Impact Statement

This work contributes to the responsible deployment of vision-language models by enabling more efficient and adaptive test-time compute allocation. By identifying when explicit reasoning helps versus when direct perception suffices, our methods can reduce unnecessary computation and associated energy costs while maintaining or improving accuracy. The ability to steer models toward perception-centric or reasoning-centric behavior offers practitioners fine-grained control without retraining, potentially democratizing access to high-quality VLM inference by reducing token budgets where appropriate.

However, causal head interventions introduce new risks: malicious actors could potentially manipulate model behavior in unintended ways, or over-reliance on automated steering could mask underlying model failures. We therefore view head-level interventions primarily as white-box scientific tools for causal analysis, not as unconditional deployment mechanisms. We emphasize that our head-scaling interventions should be validated on task-specific held-out data before production deployment, and that interpretability methods, including ours, provide correlational insights that require empirical verification of their causal claims across diverse contexts. Future work should investigate robustness to adversarial inputs and establish safety guidelines for mechanistic interventions in deployed systems.

## Acknowledgements

This research was funded, in part, by the U.S. Government under ARPA-H contract 1AY2AX000062 and the Center of Excellence in Data Science and Artificial Intelligence, an Empire State Development-designated Center of Excellence. The views and conclusions contained in this document are those of the authors and should not be interpreted as representing the official policies, either expressed or implied, of the U.S. Government.

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
