# OpenReview forum: "Does Reasoning Improve Seeing? Understanding When Vision-Language Models Benefit from Thinking"
_ICML.cc/2026/Conference — ICML 2026 regular_

### Official Review · Reviewer_gavH · 2026-03-09

**Soundness:** 2
**Presentation:** 3
**Significance:** 2
**Originality:** 3
**Overall Recommendation:** 4
**Confidence:** 3

**Summary:**

This paper focuses on the concept of meta-cognitive signals within Vision-Language Models (VLMs) to determine when explicit reasoning (Thinking mode) is more beneficial than direct answering (Instruct mode). Using a multi-pass labeling scheme, the authors define targets for reasoning helpfulness and desired generation length, demonstrating these are more linearly decodable from last-token embeddings in reasoning VLMs than in instruct variants.  Via head-wise attribution and causal scaling interventions, they identify partially disentangled attention-head circuits for length control and for reasoning. The authors also demonstrate causal steering via head-scaling interventions that adjust output length and shift model performance between perception and reasoning. They report accuracy gains of up to 5.3% on reasoning tasks.

**Compliance With Llm Reviewing Policy:**

Affirmed.

**Final Justification:**

I find the problem formulation of this work to be relatively novel and of practical value. The paper presents comprehensive experiments, combining probing, head attribution, and causal intervention.

My initial concerns were mainly about the simplified probing setup, the potential noise in labels derived from limited sampling, the limited validation across model scales, and whether the identified heads should truly be interpreted as “multimodal control circuits.” The rebuttal addressed these points meaningfully by adding sensitivity analysis, 32B-scale results, and committing to stronger error analysis in the final version.

Most importantly, the authors’ follow-up response added a text-only control experiment on a reasoning-heavy MMLU subset. I find this additional evidence useful: although some coarse layerwise patterns are shared across multimodal and text-only settings, the exact head identities do not transfer, and the intervention effects are clearly weaker in the text-only case. This supports a more careful interpretation that these heads are not purely generic text-reasoning components.

Overall, the additional evidence and the authors’ revised framing have increased my confidence in the work. I therefore view the paper more positively than in my initial review and will raise my rating to 4. I encourage the authors to incorporate these new experiments and the refined interpretation into the final version.

**Key Questions For Authors:**

1. How sensitive are r_help and l_opt labels to K and decoding hyperparameters (temperature, top-p)? Is the label stable under different K values (e.g., 20, 30) and decoding strategy?
2. Could the authors provide a more systematic error analysis of linear probes? Given the inherent noise in probing techniques, a rigorous error analysis is necessary to ensure the scientific validity of a systematic analytical work.
3. Are the identified attention heads truly "multimodal control circuits" learned during VLM alignment, or simply "general text reasoning circuits" inherited from the base LLM? Since the analysis is strictly conducted on multimodal datasets, did the authors attempt to run the same probing and attribution pipeline on a text-only reasoning benchmark (e.g., GSM8K or purely textual MMLU)?

**Limitations:**

yes

**Strengths And Weaknesses:**

# Strength

1. Novel Problem Definition and Framework: The paper introduces a novel and practical framework for adaptive VLM inference by formalizing reasoning helpfulness and desired generation length as explicit meta-cognitive control targets.
2. The paper presents extensive and diverse experiments, including multiple mechanistic analysis experiments on various benchmarks, and the overall content is detailed.
3. The paper presents the key findings in a very engaging way (many figures, tables, and other visuals).

# Weakness

1. The probing strategy relies exclusively on the last token of the final layer. While this position aggregates prior context, it is somewhat simplistic. However, since many experiments in paper are based on this, trying more probes might be more rigorous. For example, the current setup may overlook the dynamic, intermediate steps inherent in multi-step multimodal reasoning.
2. The ground-truth labels used to train the probes are derived from the accuracy of multiple sampled generations (K=10). Statistically, K=10 is a relatively small sample size, which inevitably introduces variance and noise. Consequently, the downstream results (e.g. probe, head attribution rankings, and causal interventions) might be highly sensitive to this configuration, potentially undermining the overall soundness of the paper.
3. The experiments are validated only on smaller-scale models (4B and 8B). The generalizability of the conclusions to larger models remains questionable. Furthermore, the authors claim that these meta-cognitive signals are more linearly decodable in larger models. But models of only two sizes were insufficient to support the credibility of this finding.

---

> ### Author Rebuttal · Authors · 2026-03-31
>
> ### 1. use of the final-layer last-token
> Prior work has shown that final-token representations support linear decoding of internal difficulty signals, while complementary work also shows that internal states during generation encode reasoning credibility. We therefore use the last token not because earlier tokens are unimportant, but because it provides a standardized and conservative probe interface for comparing Instruct vs. Thinking representations under identical supervision. We will clarify this and discuss token-wise / layer-wise probing.
>
> ### 2. Are the labels noisy?
>
> We agree that the two labels $r_{\text{help}}$ and $\ell_{\text{opt}}$ are multi-pass estimates. We will revise the wording to make this explicit.
>
> At the same time, we would like to clarify two points:
> 1. The difficulty label is not generated by our procedure; it comes from the dataset annotation and is human-labeled.
> 2. Most prior VLM evaluators use 1 pass, and even recent multi-pass methods such as CombiGraph-Vis [1] and MIRA [2] use only 8 passes. This makes our $K=10$ setting a practical but already conservative choice.
>
> We also did an additional sensitivity analysis on a QWEN-VL-4B model with $K \in \{5,7,10,20,30,40\}$ while keeping decoding hyperparameters fixed.
> We did not vary temperature/top-$p$ because changing them can break reliable thinking-mode generation in reasoning VLMs[3], making comparisons less controlled.
> The results show that both targets stabilize quickly, with most gains already reached around 7-10 passes:
>
> | $K$ | $\mathrm{Corr}(r_K, r_{40}) \uparrow$ | $\mathrm{MAE}(\ell_{\mathrm{opt},K}, \ell_{\mathrm{opt},40}) \downarrow$ |
> |---:|---:|---:|
> | 5  | 0.885 | 63 |
> | 7  | 0.935 | 38 |
> | 10 | 0.962 | 25 |
> | 20 | 0.984 | 10 |
> | 30 | 0.971 | 5 |
> | 40 | 0.985 | 2 |
>
> We will add this table and discussion in the revision.
> ### 3. More systematic error analysis?
>
> Our goal is not merely to report a single probe fit, but to understand how reasoning affects VLM internal representations and behavior. We appreciate this suggestion and will strengthen the revision by adding: (i) mean/std over multiple random train/validation/test splits, (ii) bootstrap confidence intervals for MSE / Spearman $\rho$ / MAE, and (iii) head-ranking stability when labels are recomputed with different $K$.
>
> More importantly, our conclusions do not rely on probes alone. The probe results are consistent with head attribution, which localizes length-related vs. reasoning/difficulty-related heads, and with causal interventions, which steer output length and reasoning/perception behavior in the predicted direction. Therefore, our core claims are supported by a convergent combination of probing, localization, and causal steering, rather than by a single probe configuration.
>
> ### 4. Larger scale comparison?
>
> We have additionally evaluated Qwen3-VL-32B and observe the same qualitative trend as in the 4B/8B models. (InternVL does not provide a dense 32B variant) The 32B results further support that Thinking representations are more linearly decodable than Instruct representations.
>
> | Base model | $r_{\text{help}}$ Inst MSE $\downarrow$ | $r_{\text{help}}$ Inst $\rho \uparrow$ | $r_{\text{help}}$ Think MSE $\downarrow$ | $r_{\text{help}}$ Think $\rho \uparrow$ | $\ell_{\text{opt}}$ Inst MSE $\downarrow$ | $\ell_{\text{opt}}$ Inst MAE $\downarrow$ | $\ell_{\text{opt}}$ Think MSE $\downarrow$ | $\ell_{\text{opt}}$ Think MAE $\downarrow$ |
> |---|---:|---:|---:|---:|---:|---:|---:|---:|
> | InternVL-4B | 0.182 | 0.324 | 0.141 | 0.412 | 3247 | 42.3 | 2856 | 38.7 |
> | InternVL-8B | 0.168 | 0.341 | 0.119 | 0.458 | 3108 | 40.1 | 2531 | 35.2 |
> | Qwen3-VL-4B | 0.194 | 0.298 | 0.153 | 0.389 | 3512 | 45.8 | 3089 | 41.2 |
> | Qwen3-VL-8B | 0.173 | 0.328 | 0.126 | 0.447 | 3294 | 42.9 | 2647 | 36.8 |
> | **Qwen3-VL-32B** | **0.157** | **0.362** | **0.093** | **0.531** | **3012** | **39.1** | **2142** | **31.8** |
>
> We will add this table in the revision and change the claim to: within the tested families and scales, the trend persists and strengthens at 32B as well.
>
> ### 5. Are these truly multimodal control circuits?
>
> This is an important question, and we agree the current evidence should be interpreted carefully.
>
> Our claim is the identified heads are circuits that are **causally useful for adaptive control in multimodal inference tasks**. We do **not** claim that we have already isolated the unique modality-specific "true circuits" of visual reasoning.
> A matched text-only control (e.g., GSM8K / text-only MMLU) would indeed be the right next step to disentangle modality-specific vs. modality-agnostic control circuits.
>
> We will narrow the wording in the paper to reflect this distinction more precisely.
>
> [1]: CombiGraph-Vis: A Curated Multimodal Olympiad Benchmark for Discrete Mathematical Reasoning*.
>
> [2]: When Visualizing is the First Step to Reasoning: MIRA, a Benchmark for Visual Chain-of-Thought
>
> [3]: Wait, Wait, Wait… Why Do Reasoning Models Loop?

---

> > ### Author Rebuttal · Reviewer_gavH · 2026-04-02
> >
> > I appreciate the authors' detailed rebuttal and the additional experimental results provided. However, the interpretation of the neural circuits and the probes may still not be thorough enough.

---

> > > ### Author Response · Authors · 2026-04-07
> > >
> > > We thank the reviewer for the follow-up.
> > >
> > > To probe whether the identified heads are specific to multimodal inference, we ran the same probe + attribution + intervention pipeline on a text-only MMLU subset consisting of `formal_logic`, `logical_fallacies`, `abstract_algebra`, `college_mathematics`, `conceptual_physics`, and `high_school_statistics`, using Qwen3-VL-4B and Qwen3-VL-8B. We selected these subjects because they are relatively reasoning-intensive, straightforward to evaluate with our existing pipeline while still covering diverse forms of textual reasoning. Due to rebuttal-time limits, we restricted this additional analysis to this subset and these two representative models; we will expand it in the revision and appendix.
> > >
> > > This control sharpens our interpretation in an important way. In text-only MMLU, we do not recover the same exact head identities as on MMMU. However, we do observe a similar coarse layerwise pattern: reasoning-related heads remain concentrated in the upper layers, while length-related heads occupy a middle-layer band, although this band is broader in text-only MMLU than in MMMU. At the same time, the head-level overlap with MMMU is negligible, and matched-strength interventions have substantially weaker behavioral effects in the text-only setting.
> > >
> > >  | Model       | Dataset   | Length-head layers | Reasoning-head layers | Length control ΔAcc | Low-layer reasoning ΔAcc |
> > >  | ----------- | --------- | -----------------: | --------------------: | ------------------: | -----------------------: |
> > >  | Qwen3-VL-4B | MMMU      |              22–24 |                 35–36 |               -1.5% |                    +2.8% |
> > >  | Qwen3-VL-4B | MMLU-text |              21–27 |                 34–36 |               -0.6% |                    +1.2% |
> > >  | Qwen3-VL-8B | MMMU      |              24–25 |                 35–36 |               -0.9% |                    +3.6% |
> > >  | Qwen3-VL-8B | MMLU-text |              19–27 |                 34–36 |               -0.4% |                    +1.3% |
> > >
> > > We interpret these results cautiously. They suggest that what transfers across domains is primarily a coarse functional pattern across layers, rather than a fixed set of identical heads. This argues against the strongest version of the “these are simply the same generic text-reasoning heads” interpretation, since the exact head identities do not transfer and their causal impact is markedly weaker in text-only reasoning. At the same time, because some layerwise structure is shared, we do not claim that these heads are exclusively multimodal or uniquely induced by visual alignment.
> > >
> > > Our revised claim is therefore narrower: the identified heads are causally useful control components for adaptive multimodal inference, while the new text-only control suggests that they may build on a partially shared, modality-general patternal scaffold whose behavioral importance becomes stronger in the multimodal setting. We will revise the paper to make this distinction explicit.

---

### Official Review · Reviewer_YM6p · 2026-03-11

**Soundness:** 3
**Presentation:** 3
**Significance:** 3
**Originality:** 4
**Overall Recommendation:** 4
**Confidence:** 3

**Summary:**

The paper investigates whether modern VLMs internally know when to use explicit reasoning versus when to just answer directly. The authors conduct three main experiments on the InternVL and Qwen3-VL model families: 1) probing meta cognitive signals, 2) localizing circuits, 3) causal interventions.

**Compliance With Llm Reviewing Policy:**

Affirmed.

**Final Justification:**

Most of my concerns are resolved. Thank you!

**Key Questions For Authors:**

Please see the weaknesses section

**Limitations:**

Yes

**Strengths And Weaknesses:**

> Strengths

- The paper addresses an important and underexplored question: reasoning is not always beneficial for multimodal reasoning. Understanding when to reason is important for efficient inference.
- This paper successfully separates the model's internal token-budget control from its difficulty assessment, offering valuable insights into how VLMs allocate compute.

> Weaknesses

- Experiments are conducted on only two VLM families with 4B and 8B parameter sizes. As an analysis paper, it would be better to evaluate on different models and larger model sizes to prove its scalability
- While circuits are identified empirically, the paper does not provide a deeper theoretical explanation of why reasoning training produces these structures. It would be helpful to also provide such a discussion.
- Although the causal interventions improve accuracy, the improvement (~5%) is relatively small and may not justify the additional complexity. In addition, as mentioned, exposing these localized causal circuits introduces additional vulnerability.

---

> ### Author Rebuttal · Authors · 2026-03-31
>
> We thank the reviewer for the thoughtful and constructive feedback. We address each concern below.
>
> ### 1. Scalability to larger models
>
> We agree that broader scale coverage strengthens the paper. To address this point directly, we added a **32B model from the same family, Qwen3-VL-32B** (InternVL does not provide a dense 32B variant), and found that the same trend not only persists but becomes stronger at larger scale.
>
> The new 32B result is shown below.
>
> | Base model       | rhelp Inst MSE ↓ | rhelp Inst ρ ↑ | rhelp Think MSE ↓ | rhelp Think ρ ↑ | ℓopt Inst MSE ↓ | ℓopt Inst MAE ↓ | ℓopt Think MSE ↓ | ℓopt Think MAE ↓ |
> | ---------------- | ---------------: | -------------: | ----------------: | --------------: | --------------: | --------------: | ---------------: | ---------------: |
> | InternVL-4B      |            0.182 |          0.324 |             0.141 |           0.412 |            3247 |            42.3 |             2856 |             38.7 |
> | InternVL-8B      |            0.168 |          0.341 |             0.119 |           0.458 |            3108 |            40.1 |             2531 |             35.2 |
> | Qwen3-VL-4B      |            0.194 |          0.298 |             0.153 |           0.389 |            3512 |            45.8 |             3089 |             41.2 |
> | Qwen3-VL-8B      |            0.173 |          0.328 |             0.126 |           0.447 |            3294 |            42.9 |             2647 |             36.8 |
> | **Qwen3-VL-32B** |        **0.157** |      **0.362** |         **0.093** |       **0.531** |        **3012** |        **39.1** |         **2142** |         **31.8** |
>
> This added result strengthens our main claim: as scale increases, the model’s internal meta-cognitive signals become more explicit, especially in the Thinking variant. Concretely, compared with Qwen3-VL-8B, Qwen3-VL-32B improves all eight probe metrics, with the largest gain appearing in Thinking-mode rhelp prediction. We will add this table and update the discussion to make the scaling trend explicit.
>
>
> ### 2. Why does reasoning training produce these structures?
>
>
> We agree that this interpretation is consistent with recent work. Prior studies show that LLMs encode problem difficulty in structured, localizable representations [1], internally represent reasoning-step credibility in ways that can be extracted by lightweight probes [2], and can be trained to separate reasoning control from task solving [3] while also optimizing reasoning efficiency directly [4]. Together, these results support our interpretation that reasoning-oriented training does not only improve answers, but also makes signals for difficulty, reasoning helpfulness, and compute allocation more explicit and manipulable. Our contribution is to provide multimodal mechanistic evidence for this in VLMs: Thinking models show a clearer separation between lower/mid-layer length heads and higher-layer reasoning/difficulty heads, with substantial overlap between the latter two, and these signals are more linearly organized than in Instruct models.
>
>
>
> ### 3. Is a ~5% causal gain meaningful?
>
> In our setting, the intervention is intentionally lightweight: we do not retrain the model, add an external verifier, or run expensive multi-sample search. We simply rescale a small localized set of heads identified by attribution analysis. Under such a minimal intervention, the observed gains are meaningful because they serve as causal validation that the identified circuits are functionally relevant, rather than merely correlational.
>
> Also, the value of Experiment 3 is not only the absolute improvement in one scalar metric. The intervention provides two practical and scientific payoffs at once. First, scaling the length heads gives a monotonic token-budget control knob, changing generation length by roughly 25–33% with only small accuracy changes. Second, scaling reasoning heads produces a depth-dependent trade-off: lower-layer scaling improves perception-dominant cases, while higher-layer scaling improves reasoning-dominant cases, with gains up to +5.3% on MMMU and similar behavior across MathVista, ScienceQA, ChartQA, and A-OKVQA.
>
> We will revise the paper to make this framing clearer and avoid overselling the intervention as a standalone deployment method.
>
> ### 4. On vulnerability and safety
>
> We agree that exposing localized circuits can introduce additional risk, and we appreciate the reviewer for raising this. Our intent is not to advocate unconditional deployment of head-level interventions. Rather, we view them as white-box scientific tools for causal analysis. Indeed, the paper already notes that mechanistic interventions should be validated carefully and that robustness to adversarial inputs remains an open question.
>
> In the revision, we will make this limitation more explicit.
>
>
> [1] https://arxiv.org/abs/2510.05969
>
> [2] https://arxiv.org/abs/2511.06209
>
> [3] https://arxiv.org/abs/2505.13379
>
> [4] https://arxiv.org/abs/2505.11225

---

### Official Review · Reviewer_rUgJ · 2026-03-17

**Soundness:** 3
**Presentation:** 2
**Significance:** 3
**Originality:** 3
**Overall Recommendation:** 4
**Confidence:** 3

**Summary:**

The authors focus on the question of when and how long VLMs reason, proposing two oracle labels, reasoning helpfulness and desired generation length. They train a probing layer with the last-token hidden states, labeled by Instruct and Thinking modes. They then use the probing layer on all attention heads to attribute and figure out heads controlling length, difficulty, and reasoning effort. They use an intervention method on multiple benchmarks to show that scaling these heads controls length and could be tuned between perception-dense and reasoning-dense tasks.

**Compliance With Llm Reviewing Policy:**

Affirmed.

**Final Justification:**

My concerns have been addressed.

**Key Questions For Authors:**

Please clarify the items in Weaknesses. I'm willing to increase my ratings accordingly.

**Limitations:**

Complementary limitation statement: They need the tested model to have both think and instruct versions to sample data and train the probing layer. To make the probing layer more robust, they also need to sample a relatively large amount of data.

**Strengths And Weaknesses:**

Strengths:
 - This work is very complete. It cleverly uses the instruct and thinking versions of the VLMs, and stably gets two kinds of answers with different reasoning effort to train probing. Then, based on interpretability experiments, it shows that length, difficulty, and thinking heads could be probed out across different model series and sizes. It also does intervention experiments, and shows that these results exist across different benchmarks.

Weaknesses:
 - The intervention conditions and following experiments are confusing. It seems like they define Sperc but didn’t present any experiment results with reasoning increase/decrease conditions. They present results of depth-controlled reasoning steering conditions, but the definition of high/low layer reasoning head is unclear (are they differentiated by layer index?). It would be better clarified by both definition and a figure like Figure 5. Due to the vagueness, it’s hard to review if the results are solid enough in Table 5 and Table 6.
 - The meaning of the background color of each heads in Figure 5 is not clear. These three kinds of heads should have different attribute scores, but it does not explain what the background color represents.
 - In the probing experiment, even for the thinking model, the correlation scores in Table 1 are not very high.
 - The font sizes of Figure 4 are not suitable and needs to be adjusted.

---

> ### Author Rebuttal · Authors · 2026-03-31
>
> We thank the reviewer for the constructive feedback and hope the clarifications below will address the concerns.
>
> ### 1. Clarification of $S_{\text{perc}}$:
> $S_{\text{perc}}$ is derived from the perception–reasoning cohort contrast of the $r_{\text{help}}$ probe. For each head, we measure the change in predicted $r_{\text{help}}$ under head ablation in the two cohorts, compute
>    $\Delta_{\ell,i}^{(r_{\text{help}})} = A_{\ell,i}^{(\text{perc-dom})} - A_{\ell,i}^{(\text{reas-dom})}$,
>    and select the top-$K$ heads in the direction opposite to $S_{\text{reas}}$ (Secs. 5.2, 5.3, 6.2).
>    Intuitively, $S_{\text{perc}}$ captures heads whose amplification shifts the model toward more perception-grounded and less reasoning-intensive behavior.
>
> ### 2. Where is $S_{\text{perc}}$ used?
>
> The key point is that our control target is **signed helpfulness**. In Sec. 3.1, $r_{\text{help}}=\hat p_{\text{THINK}}-\hat p_{\text{INST}}$, so positive values indicate that reasoning helps, while negative values indicate that reasoning hurts. Therefore, a one-directional “increase reasoning only” intervention risks mainly increasing verbosity rather than improving correctness. As shown in the length experiment and in Sec. 6.3 / Table 4, scaling a circuit can increase output length while leaving accuracy nearly unchanged. This is why we use the push–pull formulation in Eq. 3, with $S_{\text{up}}=S_{\text{reas}}$ and $S_{\text{down}}=S_{\text{perc}}$ for reasoning increase, and the reverse for perception increase. Our goal in Sec. 6 is to test **steering along the perception–reasoning axis**, not to present $S_{\text{perc}}$ as a separate unrelated circuit. For this reason, $S_{\text{perc}}$ appears in the paper as part of the reasoning increase/decrease intervention rather than as an isolated control. We agree this should be stated more explicitly in the revision.
>
> ### 3. Definition of low/high reasoning heads.
> Yes, the low/high split is **by layer index**. In Sec. 6.2, we divide reasoning heads into $S_{\text{reas}}^{\text{low}}$ (lower layers) and $S_{\text{reas}}^{\text{high}}$ (upper layers): layers with index smaller than $ layer_{num} // 2$ belong to the low group, and the rest belong to the high group.
>
> ### 4. Clarification of Tables 5 and 6.
> Table 5 shows scaling low-layer reasoning heads helps the perception cohort, while scaling high-layer reasoning heads helps the reasoning cohort. Table 6 shows that different datasets benefit from different types of steering: for some, increasing reasoning is more helpful, while for others, increasing perception is more effective.
> ### 5. Figure 5 and Figure 4
> We apologize that Figure 5 was not sufficiently self-explanatory. Our intent was that the background heatmap visualizes the head-attribution scores used in Experiment 2, i.e., the probe-sensitivity signal obtained by measuring how the probe prediction changes when an individual head is ablated/scaled, while the green/red/blue outlines indicate the top-K head sets. In the revision, we will clarify the caption/text and add a clearer legend/color explanation.
> We will also enlarge Figure 4, increase font sizes.
>
> ### 6. Correlation values in Table 1.
>
> We agree that the absolute Spearman correlations are moderate rather than very high. However, the claim in Experiment 1 is comparative, not absolute: under the same labels, data split, and linear probe setup, the Thinking representations consistently achieve lower MSE and higher $\rho$ than the Instruct representations across all four base models. We believe the moderate correlation ceiling is expected because $r_{\text{help}}$ is itself a noisy target: it is defined as the difference between two empirical correctness rates, each estimated from only $K=10$ stochastic generations, and the probe is further restricted to be linear and to operate only on the final-layer last-token embedding.

---

> > ### Author Rebuttal · Reviewer_rUgJ · 2026-04-04
> >
> > Thank the authors for their responses.

---

> > > ### Author Response · Authors · 2026-04-07
> > >
> > > Thank you for the careful review and for the positive update!
> > > We appreciate your feedback and are glad that our rebuttal addressed your concerns.
> > > We will incorporate these clarifications and additional results into the final revision.

---

### Decision · Program_Chairs · 2026-04-30

**Decision:**

Accept (regular)

**Comment:**

**Summary:** This work explores the trade-offs in VLMs between explicit reasoning (e.g. "think mode") versus answering questions directly in VLMs. Using probing experiments, this work finds that localized circuits emerge that can internally determine when reasoning is necessary and how much computation to use.

**Strengths/Weaknesses:** Reviewers appreciate the thoroughness of the experiments, including interpretability experiments and interventions. They also appreciate that two different types of models -- thinking versus instruct -- are used to compare reasoning efforts. As a more minor strength, reviewers appreciated the results being presented clearly and in many different visuals.

The weaknesses were mostly addressed during rebuttal. Two reviewers note that the models evaluated were somewhat small ($\leq$8B) and spanned only 2 model families. In the rebuttal, the authors included a 32B model and found the trends held. While other model families would be nice, it's not necessary for this work's contribution. Another weaknesses was a lack of deeper theoretical discussion around *why* these reasoning/length circuits emerge. The authors provided a satisfactory hypothesis in their rebuttal. The other weaknesses are more minor (e.g. label colors in tables, changing font sizes).

**Recommendation:** I recommend acceptance, based on the positive reviews and the main concerns being addressed during the rebuttal.